# Silicon-Based Metastructure Optical Scattering Multiply–Accumulate Computation Chip

**DOI:** 10.3390/nano12132136

**Published:** 2022-06-21

**Authors:** Xu Liu, Xudong Zhu, Chunqing Wang, Yifan Cao, Baihang Wang, Hanwen Ou, Yizheng Wu, Qixun Mei, Jialong Zhang, Zhe Cong, Rentao Liu

**Affiliations:** 1National Research Center for Optical Sensing/Communications Integrated Networking, Department of Electronics Engineering, Southeast University, Nanjing 210096, China; xdzhu@seu.edu.cn (X.Z.); wangchunqing22@163.com (C.W.); 213190182@seu.edu.cn (Y.C.); wangbaihang96@163.com (B.W.); efonsou@seu.edu.cn (H.O.); raymond.wu601@gmail.com (Y.W.); qixun_mei@foxmail.com (Q.M.); zjl17582942649@163.com (J.Z.); 213202538@seu.edu.cn (Z.C.); 213200797@seu.edu.cn (R.L.); 2State Key Lab of Mathematical Engineering and Advanced Computing, Wuxi 214125, China

**Keywords:** optical neural network (ONN), multiply–accumulate (MAC) operation, inverse design, silicon-on-insulator (SOI), metastructure, coarse wavelength division multiplexer (CWDM), optical scattering unit (OSU)

## Abstract

Optical neural networks (ONN) have become the most promising solution to replacing electronic neural networks, which have the advantages of large bandwidth, low energy consumption, strong parallel processing ability, and super high speed. Silicon-based micro-nano integrated photonic platforms have demonstrated good compatibility with complementary metal oxide semiconductor (CMOS) processing. Therefore, without completely changing the existing silicon-based fabrication technology, optoelectronic hybrid devices or all-optical devices of better performance can be achieved on such platforms. To meet the requirements of smaller size and higher integration for silicon photonic computing, the topology of a four-channel coarse wavelength division multiplexer (CWDM) and an optical scattering unit (OSU) are inversely designed and optimized by Lumerical software. Due to the random optical power splitting ratio and incoherency, the intensities of different input signals from CWDM can be weighted and summed directly by the subsequent OSU to accomplish arbitrary multiply–accumulate (MAC) operations, therefore supplying the core foundation for scattering ONN architecture.

## 1. Introduction

With the widespread use of deep learning techniques, artificial intelligence (AI) has achieved great success in machine vision, autonomous driving, and clinical diagnosis. However, the growth of the computational capability of integrated circuit (IC) chips as a carrier for training and executing AI models is gradually slowing down and failing to meet the demand of artificial neural network (ANN). The IC chip based on the von Neumann architecture has a structural defect that cannot be avoided, as it separates the program from the data when executing operations, resulting in tidal data loads between the memory and the computational unit, which decreases the computational rate and increases the computational power consumption [1]. Researchers have addressed this problem by increasing chip integration and calculating in memory [2]. Nevertheless, for transistors, as the sizes shrink and the quantum effects become more prominent, the performances deteriorate [3]. Besides, silicon-based electronic chips also have the problem of mutual interference of electronic signals. In addition, the existing neural network algorithms are not well matched with in-memory computing [4]. All these problems eventually limit the electronics applications in AI. 

ANN based on optical devices, the so-called optical neural network (ONN), can overcome the aforementioned defects. Compared with the electronics counterpart, optical transmission inherently has the advantages of high speed, large bandwidth, low delay, high parallelism, low energy consumption, and immunity to electro-magnetic interference (EMI) [5]. Therefore, ONN can take advantage of photonic technology to break the bottleneck of conventional electrical ANN.

In 2017, Shen et al. [6] successfully developed the first ONN chip based on photonic interferometric units composed of 56 programmable Mach–Zehnder interferometer (MZI) arrays in a silicon photonic integrated circuit (PIC) to achieve linear operations on matrices. They fulfilled linear operations on a layer of fully connected neural networks and obtained 76.7% recognition accuracy in vowel recognition task using this chip twice. In 2018, Lin et al. [7] proposed an all-optical diffractive depth neural network (D^2^NN) consisting of multiple layers of diffractive surfaces that could perform computational tasks at the speed of light. The training of D^2^NN was completed by computer. After completing the training, a 3D model of each diffraction layer was established by Poisson surface reconstruction method. Then, the diffraction surface was printed by a 3D printer to classify handwritten digits and images of fashionable items. In 2020, Qu et al. [8] proposed an ONN structure based on light scattering units to carry out high-speed, small-region and low-power deep learning tasks such as high-precision random matrix multiplication with mean square error less than 10^−4^ and 97.1% recognition accuracy on the classical image classification dataset MNIST. In 2021, Zhang et al. [9] took advantage of the efficient complex-valued computation in optics and designed a novel ONN chip, realizing a true complex neural network with stronger learning capability than that of real-valued one.

Based on a silicon-on-insulator (SOI) processing platform, we propose a four-channel coarse wavelength division multiplexer (CWDM) and an optical scattering unit (OSU) chip architecture, to inversely design their inner topologies, to forwardly verify such basic metastructure elements, and finally to export mask patterns for fabrication. In this paper, CWDM is used to decompose the input broadband optical signal into four channels with different wavelengths and input them to OSU. The metastructure cascades the CWDM and OSU to implement the arbitrary multiply–accumulate (MAC) operations, realizing the core functionality of ONN.

## 2. Inverse Design Methodology

In silicon photonics, a CMOS-compatible SOI platform illustrated in Figure 1 is commonly used, where the three-layer structure is given. For the silicon rib waveguide core layer the permittivity $\varepsilon_{Si}$ is 7.84, while for the silicon-oxide cladding layer $\varepsilon_{SiO_{2}}$ is 2.07. The thickness of the rib waveguide is 220 nm while the width is 400 nm [10]. The CWDM and OSU discussed subsequently are designed and fabricated in SOI.

For optoelectronic devices, the traditional forward design cycles [11] involve systematically varying a small set of characteristic parameters, simulating these devices to determine their performance, and iterating to meet design requirements. Although this process is straightforward, even an experienced designer struggles when more than a few design parameters are to be coordinated simultaneously. To overcome human limitations, researchers employ inverse design methods to begin anticipating strong candidates and more quickly converge on a satisfactory one [12,13,14]. Molesky et al. [12] reviewed recent advancements in computational inverse design approaches, focusing on algorithms for optimizing optical structures based on desired functionalities to reshape the unconventional structures available to nanophotonics. Lin et al. [13] showed that with careful consideration and reformulation of the design objects, powerful inverse design techniques could be successfully applied to a multitude of interesting problems with rich physical behaviors to address intractable computational problems. Roques-Carmes et al. [14] proposed a framework for the inverse design of multilayer metaoptics via topology optimization and implemented a 3D-printed light concentrator working at five different nonparaxial angles of incidence, paving the way to realize multifunctional ultracompact 3D nanophotonic devices. 

While in Lumerical FDTD software (Ansys Canada Ltd., Vancouver, BC, Canada) this paradigm shift is implemented using *LumOpt* as a python module. *LumOpt* comes in two different approaches—shape and topology optimization [15]. Rather than shape optimization, topology optimization generally needs more computation time to perform a few more complicated steps to produce physically realizable devices. This numerical algorithm often scales to thousands of parameters, but based on heuristic adjoint method [16,17], we only need two simulations per iteration to calculate the gradient. Furthermore, since the permittivity values are the optimization parameters, there is a direct link to Maxwell’s equations.

Once the optimization is running it proceeds in two or three steps. First comes the greyscale phase where the parameters vary continuously between the core and cladding index. Next, a binarization phase is run to force the mesh cells to take either the core or cladding index. Finally, an optional design for manufacturing (DFM) step is implemented. In this step a penalty function is added to the figure-of-merit (FOM). The magnitude of the penalty function is determined by the degree to which the minimum feature size constraint is violated. Typically, the minimum feature size is set about 500 nm, the finest resolution of mask aligner.

## 3. Four-Channel CWDM Chip Topology Optimization

### 3.1. Inverse Design

WDM unit combines a series of different wavelength optical signals into one channel as multiplexer (MUX) and splits them into each wavelength vice versa as demultiplexer (DEMUX). CWDM distinguishes from dense wavelength division multiplexer (DWDM) by broader wavelength channel separation, typically 10–20 nm, loosens the requirements for fine tunability of light source and wavelength division, and therefore could be used simply and cost-effectively.

A four-channel wavelength DEMUX working around a 1550 nm window is depicted in Figure 2. There is one input broadband source channel in the left and also four output channels in the right with the central wavelength of 1550 nm, 1570 nm, 1590 nm, and 1610 nm, individually. For each channel, the spectral width is 10 nm and the separation is also 10 nm. Therefore, the result is four divided transmission channels around [1545 nm, 1555 nm], [1565 nm, 1575 nm], [1585 nm, 1595 nm], [1605 nm, 1615 nm] routing to Port 1, 2, 3, and 4, respectively. In the core region of 6 μm×6 μm, four S-bends connect and route lightwave from input port to output ports, accomplishing the function of CWDM-DEMUX.

We set FOMs as monitors for the four output ports demonstrating the wavelength division equality, which decrease and converge to a constant value as the iteration number increases, as shown in Figure 3a. Figure 3b depicts the gradient fields due to slight perturbation of permittivity. After optimization, the permittivity distribution and hence the irregular topology of the metastructure is illustrated in Figure 3c prototyping the DEMUX layout with GDSII format file exported for further simulation and fabrication. While Figure 3d plots the forward direction optical field distribution within the device’s structure. The computing duration lasts about 5–6 h on a Dell server configured with Intel i9-10900X CPU and 128G RAM. It also depends on the structure complexity and iteration numbers.

### 3.2. Forward Verification

To verify the functionality and performance of the inversely designed DEMUX, the optical transmission characteristics (Figure 4) and field trajectory (Figure 5) of the four-channel window are simulated in the Lumerical FDTD integrated developing environment (IDE). In Figure 4, for each central wavelength of 1550 nm, 1570 nm, 1590 nm, 1610 nm the transmittance approximates to approximately 90% while for the boundary wavelength of each channel it descends to 70–85% due to the DFM treatment of inverse design strategy, which shall be discussed in the following Section 3.3. The less than 1 dB difference for each channel’s central wavelength and boundary wavelength demonstrates the rather flat transmittance performance of the device. Figure 5a–d represents the optical field propagation within the device routing to port 1, 2, 3, and 4 for different central wavelength of 1550 nm, 1570 nm, 1590 nm, 1610 nm, respectively.

### 3.3. DFM Strategy for Mask Design

The inversely designed and subsequently forward-validated metastructure tends to create features that break process design rules. To ensure DFM, Lumerical IDE supplies ways of explicitly enforcing specific minimum feature size constraints [18]. The main idea is to come up with an indicator function which is minimal if the minimum feature size constraints are fulfilled everywhere. DFM treatment implements the manufacturability albeit the device performance degrades to some extent, which can be seen from the transmittance difference between central and boundary wavelength for each divided channel shown in Figure 4 of previous Section 3.2.

Figure 6 compares the performance-priority design and manufacturability-priority design for CWDM-DEMUX’s permittivity distribution. For the former, small holes or disconnected parts may appear in the prototype; while for the latter, due to DFM treatment, the minimum feature speckles are neglected for easy fabrication of the metastructure. Thus, when exporting pattern definition to GDSII file illustrated in Figure 7 from Figure 6b, bigger discontinuous silicon pillars or voids of irregular shapes have replaced smaller ones depicted in Figure 6a. 

## 4. Inverse Design of OSU Chip

Based on inverse design methodology, we conceive an initial guess of OSU metastructure with multi-input–multi-output (MIMO) on SOI, which has four input ports and four output ports as an instance, illustrated in Figure 8. The square scattering region is 6 μm×6 μm. The forward TE optical modes with wavelength window of $\lambda_{1}\in \left[1545\;\mathrm{nm},1555\;\mathrm{nm}\right]$, $\lambda_{2}\in \left[1565\;\mathrm{nm},1575\;\mathrm{nm}\right]$, $\lambda_{3}\in \left[1585\;\mathrm{nm},1595\;\mathrm{nm}\right]$ and $\lambda_{4}\in \left[1605\;\mathrm{nm},1615\;\mathrm{nm}\right]$, which come from the aforementioned CWDM-DEMUX device, stimulate into four input ports, respectively. For each light source, the optical intensity is *I*_1_(*λ*_1_), *I*_2_(*λ*_2_), *I*_3_(*λ*
_3_)and *I*_4_(*λ*_4_) accordingly. Through the OSU, i.e., the MIMO system, we obtain the mixed and redistributed optical signal at output ports with the intensity of *I*_*o*1_, *I*_*o*2_, *I*_*o*3_and *I*_*o*4_ The relationship between the input optical intensity *I_in_* and output optical intensity *I_out_* is given by Equations (1) and (2). Due to the light source incoherency and scattering behavior, the intensities of different input signals can be weighted and summed directly so that the OSU perfectly fulfills the specific MAC computation represented by the stochastic matrix *P* defined in Equation (3). Thus, arbitrary MAC operations based on random optical power splitting ratio and summation could be accomplished by OSU.

The FOMs of the OSU can converge when the iteration number reaches 800 as Figure 9a indicates. Figure 9b depicts the sparse perturbation gradient fields. The permittivity distribution and hence the inner topology of the metastructure is illustrated in Figure 9c prototyping the OSU chip layout. While Figure 9d vividly portrays the random scattering process through photons colliding with the metastructure particles such as silicon pillars or voids.


(1)
$$I_{\mathrm{out}}{=PI}_{in}=\left[\begin{array}{cccc} 0.25 & 0 & 0 & 0\\ 0.25 & 0.33 & 0 & 0\\ 0.25 & 0.33 & 0.5 & 0\\ 0.25 & 0.33 & 0.5 & 1 \end{array}\right]\left[\begin{array}{c} I_{1}\\ I_{2}\\ I_{3}\\ I_{4} \end{array}\right]{=I}_{1}\left[\begin{array}{c} 0.25\\ 0.25\\ 0.25\\ 0.25 \end{array}\right]{+I}_{2}\left[\begin{array}{c} 0\\ 0.33\\ 0.33\\ 0.33 \end{array}\right]{+I}_{3}\left[\begin{array}{c} 0\\ 0\\ 0.5\\ 0.5 \end{array}\right]{+I}_{4}\left[\begin{array}{c} 0\\ 0\\ 0\\ 1 \end{array}\right]=\left[\begin{array}{c} I_{o1}\\ I_{o2}\\ I_{o3}\\ I_{o4} \end{array}\right]$$




(2)
$$\begin{array}{l} I_{o1}=0{.25I}_{1}{(\lambda }_{1})\\ I_{o2}=0{.25I}_{1}{(\lambda }_{1})+0{.33I}_{2}{(\lambda }_{2})\\ I_{o3}=0{.25I}_{1}{(\lambda }_{1})+0{.33I}_{2}{(\lambda }_{2})+0{.5I}_{3}{(\lambda }_{3})\\ I_{o4}=0{.25I}_{1}{(\lambda }_{1})+0{.33I}_{2}{(\lambda }_{2})+0{.5I}_{3}{(\lambda }_{3}{)+I}_{4}{(\lambda }_{4}) \end{array}$$


(3)
$$P=\left[\begin{array}{cccc} 0.25 & 0 & 0 & 0\\ 0.25 & 0.33 & 0 & 0\\ 0.25 & 0.33 & 0.5 & 0\\ 0.25 & 0.33 & 0.5 & 1 \end{array}\right]\;$$



Moreover, we can detail each optical signal propagation characteristics as follows. The optical signal *λ*_1_ is almost equally routing to the four output ports as the relative electric intensity curve at certain output port indicates in Figure 10a while the field trajectory is depicted in Figure 11a. The optical signal λ_2_ is equally routing to the output port 2, 3, and 4 as the relative electric intensity curve indicates in Figure 10b while the field trajectory is depicted in Figure 11b. The optical signal λ_3_ is equally routing to the output port 3 and 4 as Figure 10c demonstrates, while the field trajectory is depicted in Figure 11c. The optical signal λ_4_ is directly routing to the fourth output port shown in Figure 10d while the field trajectory is depicted in Figure 11d.

For the sake of DFM rule, as with CWDM-DEMUX, we export the OSU layout illustrated in Figure 12 with GDSII file for fabrication.

In the end, as depicted in Figure 13, the silicon-based scattering metastructure cascading CWDM-DEMUX (Figure 7) and OSU (Figure 12) perfectly accomplishes arbitrary MAC operations and hence supplies the sound foundation for scattering ONN architecture. 

Furthermore, metastructures over four channels, such as one by nine CWDM and nine by nine OSU, could fulfill nine optical signals’ weighting and summation operation by the stochastic matrix implemented on the random optical power splitting ratio and incoherency.

## 5. Conclusions

To tackle the obstacles confronted by electronics in AI and meanwhile meet the requirements of smaller size and higher integration for silicon photonic computing, the topologies of four-channel CWDM and OSU are inversely designed and optimized in the state-of-the-art Lumerical IDE, based on which, the scattering metastructures are constructed. Due to the random optical power splitting ratio and incoherency, the intensities of different input signals from CWDM can be weighted and summed directly by the cascaded OSU to accomplish arbitrary MAC operations, therefore supplying the core foundation for scattering ONN architecture. The ongoing research will focus on the optical performance and feasibility of the fabrication on conventional silicon material and universal photolithography/etching process to exploit the potential application of handwritten digit classification in ONN.

## Figures and Tables

**Figure 1 nanomaterials-12-02136-f001:**
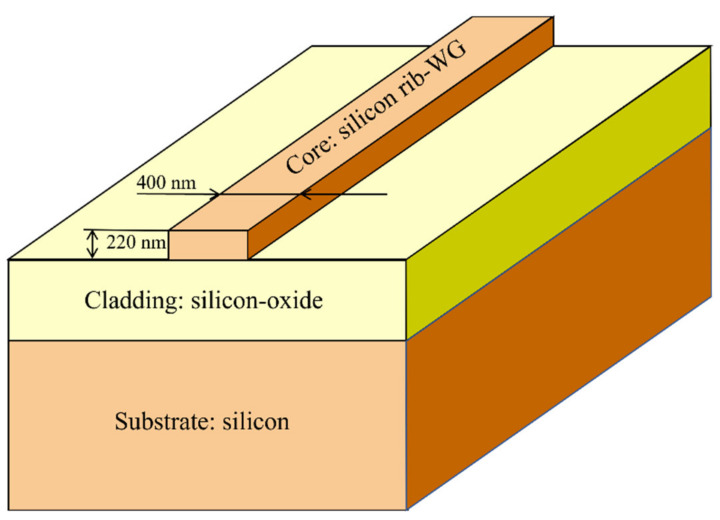
Three-layer CMOS-compatible SOI processing platform. This figure is adapted from [10].

**Figure 2 nanomaterials-12-02136-f002:**
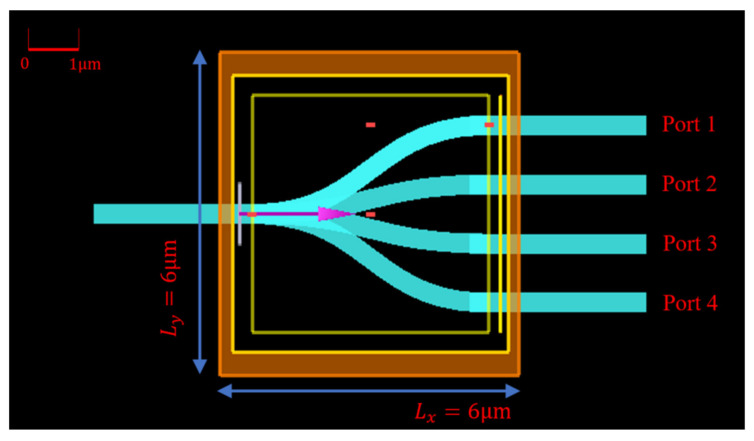
Initial guess for CWDM-DEMUX planar topology in Lumerical FDTD software.

**Figure 3 nanomaterials-12-02136-f003:**
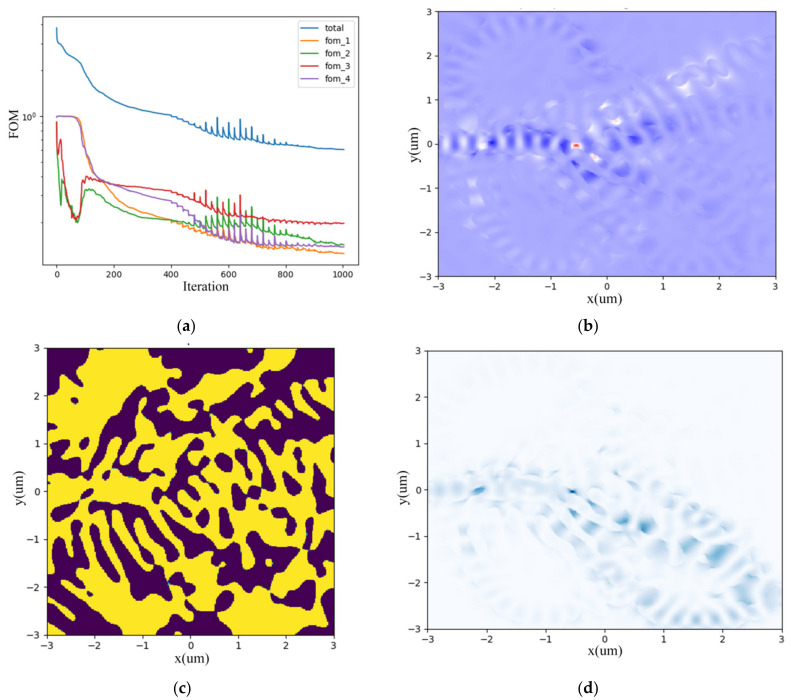
Inverse design of CWDM-DEMUX. (**a**) four-port FOM vs. iteration number; (**b**) sparse perturbation gradient field; (**c**) permittivity distribution; (**d**) forward direction field distribution within the device’s structure.

**Figure 4 nanomaterials-12-02136-f004:**
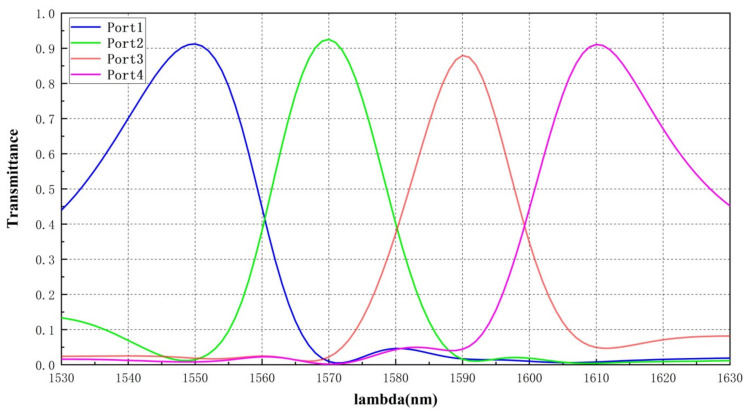
Optical transmission curves for CWDM-DEMUX.

**Figure 5 nanomaterials-12-02136-f005:**
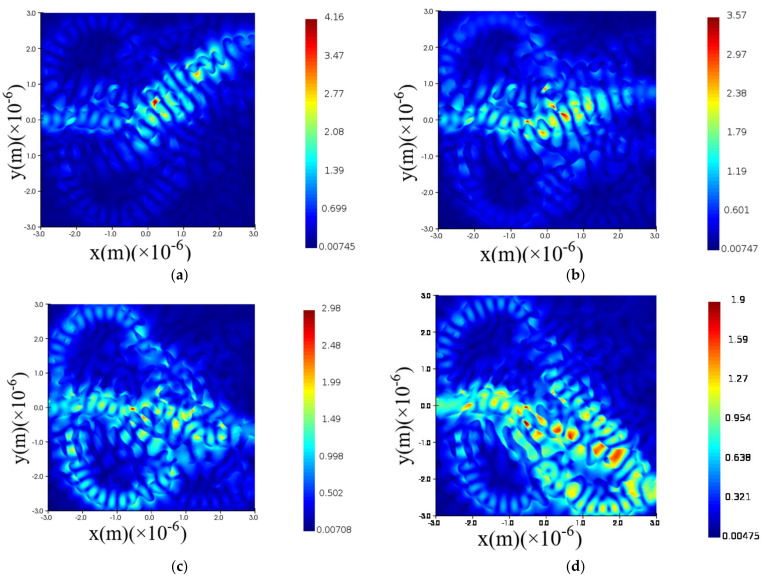
Optical field trajectory for CWDM-DEMUX at different central wavelength of (**a**) 1550 nm; (**b**) 1570 nm; (**c**) 1590 nm; (**d**) 1610 nm.

**Figure 6 nanomaterials-12-02136-f006:**
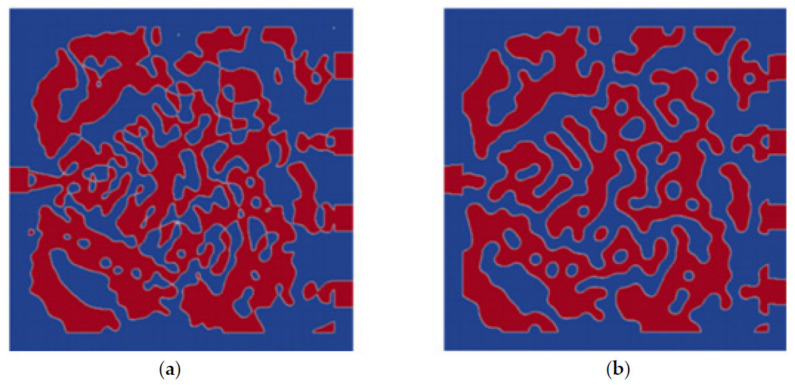
CWDM-DEMUX’s permittivity distribution for (**a**) performance-priority design in comparison with (**b**) manufacturability-priority design.

**Figure 7 nanomaterials-12-02136-f007:**
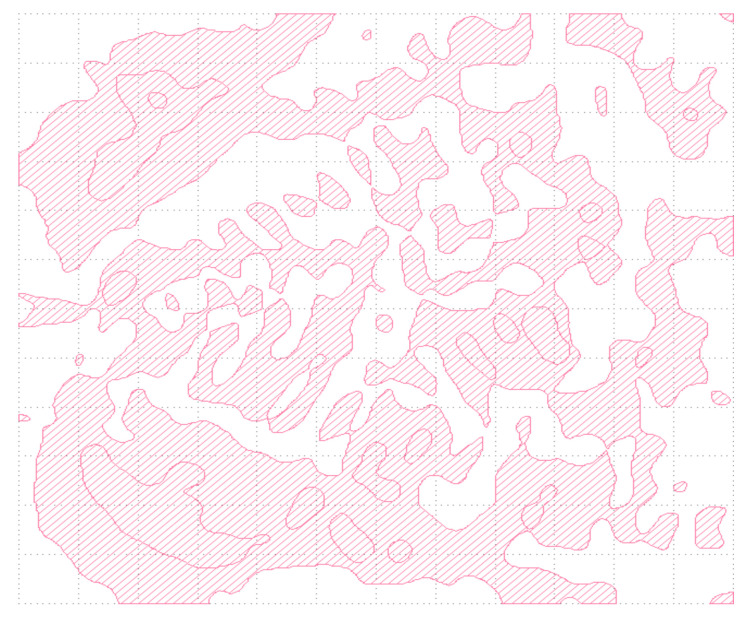
GDSII pattern definition for CWDM-DEMUX metastructure layout.

**Figure 8 nanomaterials-12-02136-f008:**
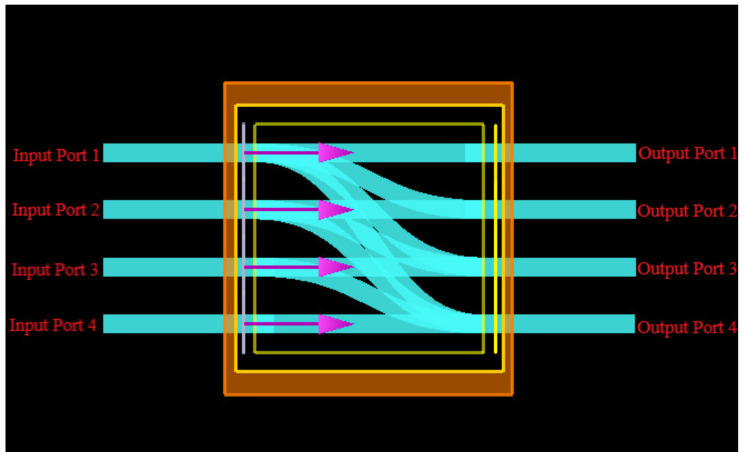
Modeling an OSU metastructure on SOI in Lumerical FDTD software.

**Figure 9 nanomaterials-12-02136-f009:**
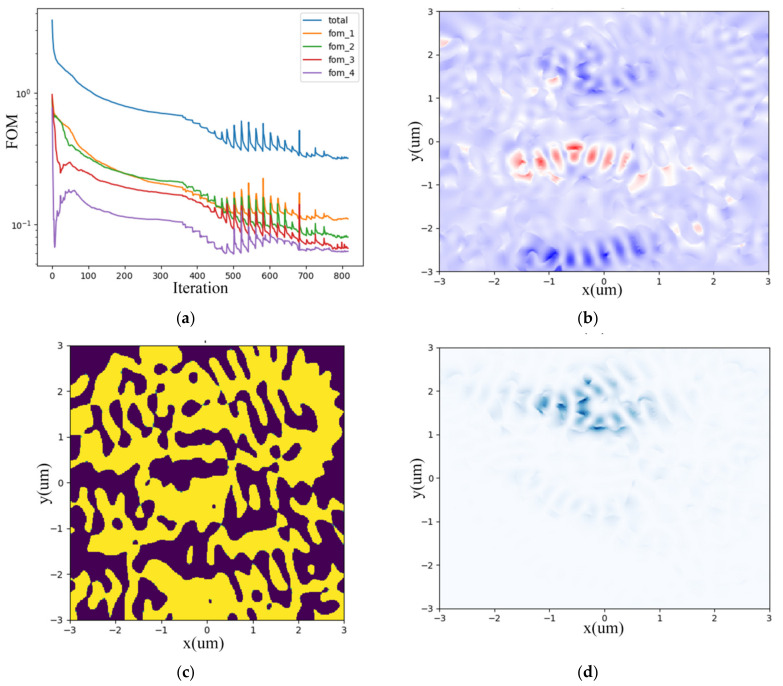
Inverse design of OSU. (**a**) four-port FOM vs. iteration number; (**b**) sparse perturbation gradient field; (**c**) permittivity distribution; (**d**) forward direction field distribution within the device’s structure.

**Figure 10 nanomaterials-12-02136-f010:**
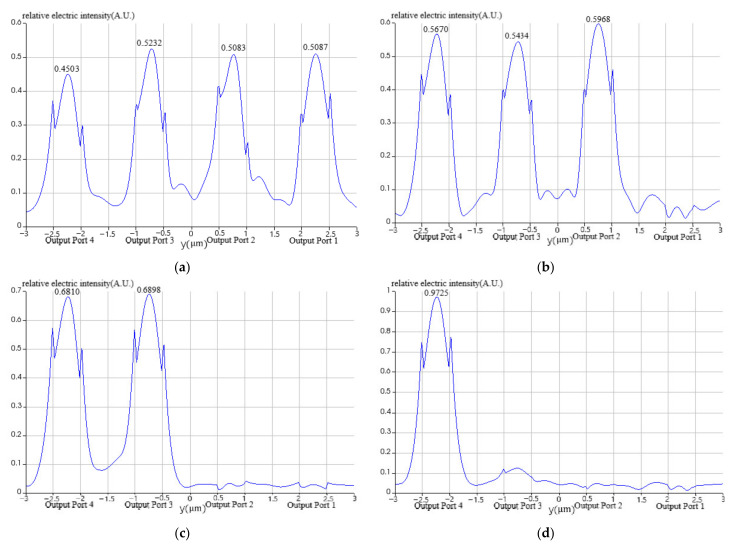
The relative electric intensity distributions at certain output port for different optical signals. (**a**) $\lambda_{1}\in \left[1545\;\mathrm{nm},1555\;\mathrm{nm}\right]$; (**b**) $\lambda_{2}\in \left[1565\;\mathrm{nm},1575\;\mathrm{nm}\right]$; (**c**) $\lambda_{3}\in \left[1585\;\mathrm{nm},1595\;\mathrm{nm}\right]$; (**d**) $\lambda_{4}\in \left[1605\;\mathrm{nm},1615\;\mathrm{nm}\right]$.

**Figure 11 nanomaterials-12-02136-f011:**
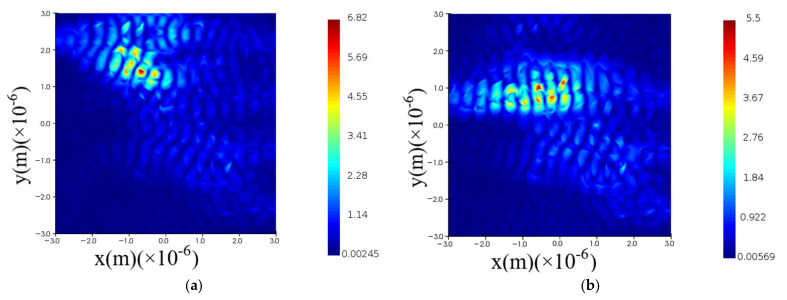
The optical field trajectory within OSU metastructure. (**a**) $\lambda_{1}\in \left[1545\;\mathrm{nm},1555\;\mathrm{nm}\right]$; (**b**) $\lambda_{2}\in \left[1565\;\mathrm{nm},1575\;\mathrm{nm}\right]$; (**c**) $\lambda_{3}\in \left[1585\;\mathrm{nm},1595\;\mathrm{nm}\right]$; (**d**) $\lambda_{4}\in \left[1605\;\mathrm{nm},1615\;\mathrm{nm}\right]$.

**Figure 12 nanomaterials-12-02136-f012:**
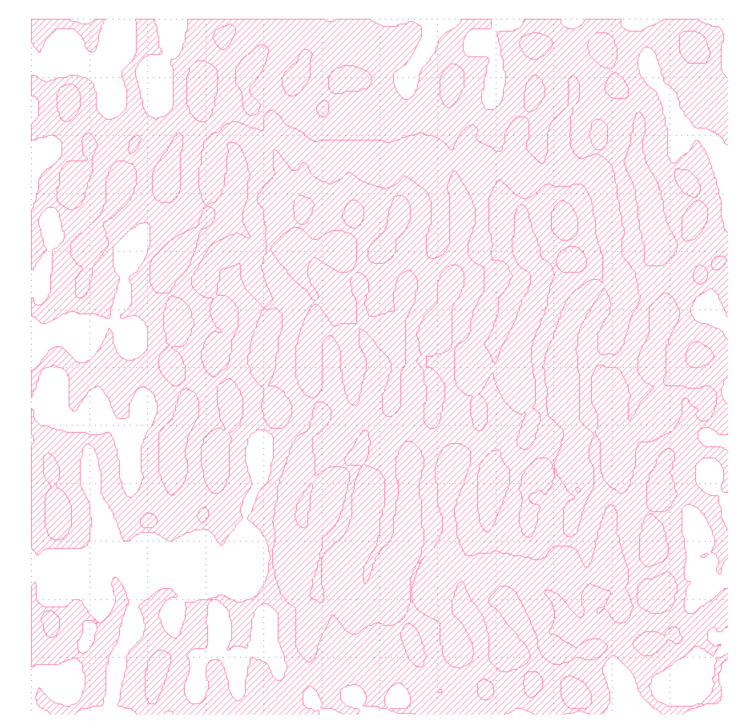
GDSII pattern definition for OSU metastructure layout.

**Figure 13 nanomaterials-12-02136-f013:**
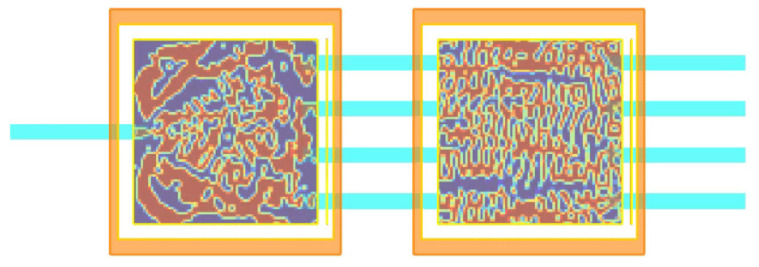
The silicon-based scattering metastructure cascading CWDM-DEMUX and OSU.

## Data Availability

Data presented in this article are available at request from the corresponding author.

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
