# Peer review of "Silicon-Based Metastructure Optical Scattering Multiply–Accumulate Computation Chip"

_nanomaterials, 2022, doi:10.3390/nano12132136_

Round 1

Reviewer 1 Report

line 10: "Optical neural network (ONN) has ..." > Please consider rephrasing "The concept of ... has ... " or " Optical neural networks (OON) have ..."

line 12: "Silicon-based micro-nano integrated photonic platform has ..." > Please consider rephrasing " ... have proven good compatibility ..." and consider a citation on this statement (at least in the text body).

lien 15/16: "can be achieved on such platform" > Consider rephrasing: "platforms"

line 36-42: "Researchers have addressed this problem by increasing chip integration and calculating in memory." > Please cite this statement and also the given examples in this paragraph.

line 68: For the reader outside silicon photonics, please write 1-2 explainatory sentences each about what a CWDM and what an OSI are (in your specific context). (Note that line 120 ff just states how CWDM distinguishes from other multiplexers).

line 80 (Figure 1): The Figure and caption (way too short) is _not_ informative in the present state. Please enhance both so that it becomes clear what is the intended message.

line 81: Please explain "traditional forward design cycle", explain which particular characteristic parameters are involved. Consider flow diagram on that issue.

line 133 (Figure 2): Please include scalebar. Please enhance the caption and mention the data type (GDSII design data? Microscope image?).

line 143 (Figure 3): Text boxes e.g. axis labels are low-res. Please improve. Diagram titles may be omitted and mentioned in the captions referring to (a), (b), (c), (d). Please enhance caption for better reading.

line 157: Please use correct spacing: 1550<blank>nm, etc.

line 159 (Figure 4): cf. Figure 3 (low res) plus fonts are way too small!

line 160/161 (Figure 5): cf. Figure 3 (low res) and cf. line 157 (blanks)

line 181 (Figure 7): cf. Figure 3 (improve low res)

line 207 (Figure 8): cf. Figure 2 (data type)

line 209 (Figure 9): increase low res, increase font size, remove header labels and include in caption

line 223 (Figure 10): dto.

line 231 (Figure 12): cf. Figure 3 (improve low res)

lines 238-245: The conclusions need to be adjusted to draw a clear line what the findings (constructed metastructures) are good for and what not (yet). It needs to answer the following questions:

  • what are potential manufacturing scenarios for each metastructure? (and what not)?
  • what are potential applications or other technical and/or societal benefits for each metastructure if manufactured (and what not)?
  • what complementary research has to be done, which brickstone walls have to be overcome that these results have an impact?

Reviewer 2 Report

Dear Sirs,

I consider the paper acceptable after minor revision.

I would like to give you my suggestions to improve the paper:

1. English is fine and understandable but can still be improved (to be clear: it'fine and I didn't find errors but it sounds a bit scholastic);

2. a description of the computing resources and of the computing time needed to optimize the metastructures would be useful and interesting to the reader;

3. concerning the optional DFM (Design For Manufacturing) optimization steps introduced in the design of both devices (DEMUX and Optical Scattering Unit) it would be extremely useful to have an indication of the minimum feature size constraints used;

4. a discussion of possible extensions to more than four channels (with expected bottlenecks and limitations) would be of general interest.

typo at line 186: light scatting region

Best regards

Reviewer 3 Report

In this manuscript, the authors report the inverse design of a 1 by 4 WDM and a 4 by 4 OSU for photonic MAC operations. The idea is generally interesting. I would only recommend the publication of this manuscript after addressing the following questions:

Major concerns:

1.      There are insufficient references to support many of the major claims in the introduction.

2.      In Fig.6 the generated designs with performance-priority or manufacturability-priority are different. How are the corresponding optical performance varied? Will Fig.6b design give much worse optical performance?

3.      What are the y-axis in Fig.10? It should be given exactly instead of a.u.. In Fig.10a, it seems the power is extremely low in Port 1, which does not seem to be acceptable.

4.      What is the insertion loss of OSU?

5.      Computing results should be simulated using the structure in Fig.13. How are the simulated computing results compared to theoretical results calculated by Equation 1?

6.      How long does it required to complete the optimization of WDM and OSU respectively?

7.      The state-of-the-art commercial WDM can have a narrow bandwidth of 0.8nm for each channel. However, in this inverse design, the bandwidth is 10nm, which will significantly limit the overall system bandwidth. What is the advantage or benefit of the inverse design then?

Minor concerns:

1.      The spectral width of each WDM channel is 10nm. But shouldn’t the separation be 20nm instead of 10nm?

2.      The respective wavelength of Fig.3b and 3d should be given.

3.      The color bars in Fig.5 should be the same for direct comparison.

Round 2

Reviewer 1 Report

Chapter 2: The chapter inverse optical design has improved, it should yet compare with the state of the art in nanophotonics (Molesky, R. E. Christiansen, Z. Liu, Lumera) which is compeletely missing. Writing shall highlight particularly the own contributions beyond state of the art.

In the conclusion, reference to this state of the art and the own contributions shall be given.

Figure 3, 7, 9: Captions shall be operational ("GDSII image of ... as an initial guess of ... ") and more enhanced to comprehend the context without considering the text.

In the Acknowledgements, in scientific papers funding sources and grant givers that supported the work shall be mentioned which is important for the reviewers and the scientific community as a whole. This is to transparently communicate certain potential biases. This is completely missing. Who paid the salaries during the project period? Additionally, people with technical contributions to the paper can me mentioned. This is maybe not necessary looking at the comprehensive author list. Pointing out the occasion of the university jubilee seems ok to me, however the personal mentioning to family members in spite of the sad occasion, is uncommon in scientific paper acknowledgements (other than in academic theses).

Please reconsider a reformulation.

Reviewer 3 Report

The concerns are not addressed well. I would not recommend the publication of this manuscript in its current status:

Response 2: Please show the optical performance similar to Fig.5 for both designs shown in Fig.7a and Fig.7b to prove the performances are comparable.

Response 3: The low transmission in Port 1 is still not acceptable. [0.25 0.25 0.25 0.25] is not really achieved. Using traditional directional couplers or MMIs, equal transmissions can be achieved in all channels independent of the relative location of output channels. Claiming the low transmission in channel 4 is caused by its long distance from the input channel is not acceptable.

Response 4: So what is the scattering loss? Insertion loss is still a key that cannot be neglected. It should not be difficult to simulate the input power and output power from the OSU. Then the overall loss is known. 

Response 5: Phase 1+Phase 2 is a complete project. Project 1 itself does not validate the idea of computing.

Response 7: For computing, the bandwidth is still important. It determines the data rate (this is where photonics will outperform electronics) and also determines the possible computation parallelism. It should be discussed how to make each channel narrower.

Since the advantage of the proposed method is 'quickly converge', what is the performance penalty then? It might not make sense if the simulation time is much faster but the performance is greatly degraded.

Round 3

Reviewer 3 Report

The authors have very well addressed my concerns except the intensity balance of [0.25 0.25 0.25 0.25]. The intensity in the first channel is 0.4226^2 = 0.17. And the second channel is 0.5491^2=0.3. This is still not acceptable as the difference is almost twice. What would happen if OSUs are cascaded in real applications? The error will soon be terrible after few layers. 

Despite this serious problem, I would recommend the publication of this manuscript.
